 eLIFE

# Serum amyloid A is a retinol binding protein that transports retinol during bacterial infection

**Mehabaw G Derebe[1†], Clare M Zlatkov[1†], Sureka Gattu[1], Kelly A Ruhn[1], Shipra Vaishnava[1], Gretchen E Diehl[2], John B MacMillan[3], Noelle S Williams[3], Lora V Hooper[1,4*]**

[1]Department of Immunology, University of Texas Southwestern Medical Center, Dallas, United States; [2]Molecular Pathogenesis Program, The Kimmel Center for Biology and Medicine of the Skirball Institute, New York University School of Medicine, New York, United States; [3]Department of Biochemistry, University of Texas Southwestern Medical Center, Dallas, United States; [4]Howard Hughes Medical Institute, University of Texas Southwestern Medical Center, Dallas, United States

**Abstract** Retinol plays a vital role in the immune response to infection, yet proteins that mediate retinol transport during infection have not been identified. Serum amyloid A (SAA) proteins are strongly induced in the liver by systemic infection and in the intestine by bacterial colonization, but their exact functions remain unclear. Here we show that mouse and human SAAs are retinol binding proteins. Mouse and human SAAs bound retinol with nanomolar affinity, were associated with retinol in vivo, and limited the bacterial burden in tissues after acute infection. We determined the crystal structure of mouse SAA3 at a resolution of 2 Å, finding that it forms a tetramer with a hydrophobic binding pocket that can accommodate retinol. Our results thus identify SAAs as a family of microbe-inducible retinol binding proteins, reveal a unique protein architecture involved in retinol binding, and suggest how retinol is circulated during infection.

*For correspondence: lora.hooper@utsouthwestern.edu

†These authors contributed equally to this work

**Competing interests:** The authors declare that no competing interests exist.

**Reviewing editor**: Fiona M Powrie, Oxford University, United Kingdom

## Introduction

Retinol plays a vital role in the physiological response to microbial challenge. Retinol is derived from dietary vitamin A and can be converted enzymatically to retinoic acid, which complexes with nuclear receptors to regulate gene transcription programs in cells (*Germain et al., 2006*). In this way, retinol promotes the maturation of innate immune cells (*Lawson and Berliner, 1999*; *Stephensen, 2001*; *Spencer et al., 2014*), governs the differentiation of adaptive immune cells (*Mucida et al., 2007*; *Hall et al., 2011*), and facilitates the regeneration of epithelial barriers damaged by infection (*Osanai et al., 2007*). A hallmark of vitamin A deficiency in humans is a markedly increased susceptibility to infection (*Sommer 2008*, *Underwood, 2004*), underscoring the broad impact of retinol on immunity.

As a small lipid-soluble compound, retinol cannot freely circulate but is instead transported among cells and tissues by specialized retinol binding proteins. Serum retinol binding protein (RBP) facilitates transport of retinol among the intestine, which is the site of retinol acquisition, the liver, which is the major site of retinoid storage, and other tissues that require retinol for their physiological functions (*Blaner, 1989*). Despite the increased requirement for retinol, serum RBP is markedly reduced following microbial challenge (*Rosales et al., 1996*), leaving open the question of how retinol is transported among tissues during infection.

Serum amyloid A (SAA) proteins are a family of proteins that are expressed in the intestinal epithelium (*Eckhardt et al., 2010*; *Reigstad and Bäckhed, 2010*) and liver (*Uhlar and Whitehead, 1999*)

**eLife digest** Vitamins are nutrients that organisms require in order to survive and grow. If an organism is unable to synthesize a vitamin in sufficient quantities, it is essential that it obtain the vitamin through its diet instead.

Vitamin A is found in foods such as eggs, animal liver and carrots, and a diet that is lacking in this vitamin can cause blindness and an increased risk of microbial infections. Vitamin A is not a single compound, but rather a collection of compounds with similar molecular structures. One of these is retinol, which plays a vital role in the body's response to microbial infection. Retinol must bind to specific proteins to be able to move through the bloodstream and be transported around the body.

Serum retinol binding protein transports ingested retinol from the intestine to the liver and other tissues. However, during microbial infection—when retinol transport is particularly important—the amount of this protein dramatically decreases; as such it is unclear how retinol is transported when the body is under attack from pathogens.

It had been suggested that Serum Amyloid A (SAA) proteins, a family of proteins made by some liver and intestinal cells, could be involved in the response to infection, because these proteins' levels increase during infection. However, their exact functions were unknown. Derebe, Zlatkov et al. found that mice fed a diet poor in vitamin A produced fewer SAA proteins in their liver and intestinal cells. However, treating the cells with retinol or the molecule it is broken down into—called retinoic acid—caused more SAAs to be made. Derebe, Zlatkov et al. also discovered that SAAs are associated with retinol in blood samples taken from mice infected with salmonella; and that both mouse and human SAAs bind tightly to retinol. Combined, this evidence suggests that SAAs are the retinol binding proteins that transport retinol during infections.

Derebe, Zlatkov et al. went on to solve the crystal structure of a mouse SAA protein, and showed that four SAA molecules bind together to form a 'pocket' that can hold a retinol molecule. Future work will focus on understanding exactly how the transport of retinol by SAAs affects the development of immunity to infections.

and circulate in the serum (*Whitehead et al., 1992*). SAA family members are encoded in the genomes of virtually all vertebrates and are highly conserved among species, suggesting essential biological functions (*Uhlar et al., 1994*). Expression of SAAs is strongly induced by microbial exposure. SAAs are induced in intestinal epithelial cells by the microbiota (*Ivanov et al., 2009*; *Reigstad et al., 2009*; *Eckhardt et al., 2010*; *Reigstad and Bäckhed, 2010*), and have been implicated in promoting Th17 cell development in response to specific microbiota components (*Ivanov et al., 2009*). Similarly, liver and serum SAAs are markedly elevated following systemic bacterial or viral infection (*Meek and Benditt, 1986*; *Chiba et al., 2009*).

Although it has been proposed that SAAs generally contribute to inflammation and immunity (*Eckhardt et al., 2010*), the exact functions of SAAs remain poorly defined. Interestingly, SAAs have characteristics that suggest they could bind hydrophobic ligands. First, all SAAs are predicted to form amphipathic helices with a hydrophobic face that could interact with non-polar molecules (*Stevens, 2004*). Second, SAAs circulate in the serum associated with high-density lipoprotein (HDL), which transports lipid-bound lipoproteins amongst tissues (*Whitehead et al., 1992*). However, the identity of potential SAA ligand(s) remains unclear.

Here, we show that mouse and human SAAs are retinol binding proteins. We demonstrate that SAA expression in mice requires dietary vitamin A, that mouse and human SAAs bind tightly to retinol, and that SAA recovered from serum following bacterial infection is associated with retinol. We find that $Saa1/2^{-/-}$ mice, which harbor deletions of both the $Saa1$ and $Saa2$ genes, have higher bacterial burdens in spleen and liver following an acute bacterial infection, supporting an essential role for SAAs in the response to microbial challenge. Finally, we provide structural insight into the binding interaction by solving the mouse SAA3 crystal structure, which reveals a tetrameric assembly with a hydrophobic binding pocket that can accommodate retinol. These studies thus identify SAAs as a family of retinol binding proteins and reveal a new protein architecture supporting retinol binding. Our findings suggest that SAAs mediate retinol transport during microbial challenge and thus constitute a key component of the physiological response to infection.

## Results

### SAA expression requires dietary vitamin A

Initially we uncovered a relationship between SAA expression and dietary vitamin A status in mice. Microarray analysis disclosed that mice fed a vitamin A-deficient diet exhibited lower abundances of serum amyloid A (*Saa*) 1 and 2 transcripts in the intestine as compared to mice fed a vitamin A-replete diet (*Figure 1—figure supplement 1*; *Table 1*). Real-time quantitative PCR and immunofluorescence analysis verified that expression of small intestinal SAA1, 2, and 3 was reduced in mice fed a vitamin A-deficient diet (*Figure 1A,B*; *Table 1*). Liver expression of SAA1 and 2 was also reduced in mice fed a vitamin A-deficient diet, although the reduction in expression was less pronounced than in the intestine (*Figure 1C,D*). This is likely because dietary vitamin A deficiency does not completely deplete stored retinoids in the liver (*Liu and Gudas, 2005*). We also observed elevated expression of intestinal *Saa1* and *Saa2* following addition of retinol directly to the epithelial surface of small intestinal explants, and of liver *Saa1* and *Saa2* after intraperitoneal supplementation with retinoic acid (*Figure 1—figure supplement 2*). These findings support the idea that retinoids directly impact *Saa* expression. Addition of retinol or retinoic acid to cultured HepG2 cells (a human liver cell line) enhanced expression of *SAA1* and *2* in the presence of IL-1β and IL-6 (*Figure 1E,F*), suggesting that the impact of dietary vitamin A on SAA expression is due to cell-intrinsic effects of retinoids. Collectively, these results show that full expression of SAAs in the intestine and liver requires dietary vitamin A.

**Table 1.** Primers used in Q-PCR analysis

| Primer name | Primer sequence |
| --- | --- |
| mouse SAA1 F | 5′-CATTTGTTCACGAGGCTTTCC |
| mouse SAA1 R | 5′-GTTTTTCCAGTTAGCTTCCTTCATGT |
| mouse SAA2 F | 5′-TGTGTATCCCACAAGGTTTCAGA |
| mouse SAA2 R | 5′-TTATTACCCTCTCCTCCTCAAGCA |
| mouse SAA3 F | 5′-CGCAGCACGAGCAGGAT |
| mouse SAA3 R | 5′-CCAGGATCAAGATGCAAAGAATG |
| human SAA1 F | 5′-GGCATACAGCCATACCATTC |
| human SAA1 R | 5′-CCTTTTGGCAGCATCATAGT |
| human SAA2 F | 5′-GCTTCCTCTTCACTCTGCTCT |
| human SAA2 R | 5′-TGCCATATCTCAGCTTCTCTG |
| mouse 18S F | 5′-CATTCGAACGTCTGCCCTATC |
| mouse 18S R | 5′-CCTGCTGCCTTCCTTGGA |
| mouse Gapdh F | 5′-TGGCAAAGTGGAGATTGTTGCC |
| mouse Gapdh R | 5′-AAGATGGTGATGGGCTTCCCG |
| human Gapdh F | 5′-CCTGGTCACCAGGGCTGCTTTTAAC |
| human Gapdh R | 5′-GTCGTTGAGGGCAATGCCAGCC |

### Human and mouse SAAs bind retinol

Transcriptional control by retinoids is frequently observed in proteins that function in retinoid transport and metabolism (*Noy, 2000*). Because SAAs have predicted hydrophobic binding surfaces (*Stevens, 2004*) and are induced by retinol, we hypothesized that they might be retinol binding proteins. We therefore tested for retinol binding activity of recombinant human SAA1 (hSAA1), mouse SAA1 (mSAA1), and mouse SAA3 (mSAA3) using fluorometric binding assays that exploit the unique spectral properties of retinol. (Note that we were unable to express recombinant mSAA2). Retinol exhibits intrinsic fluorescence that is enhanced upon binding to proteins through energy transfer from tryptophan residues, and this fluorescence change can be used to quantify binding (*Cogan et al., 1976*) (*Figure 2A*). We did fluorometric titrations to determine apparent dissociation constants ($K_d$s) for the all-*trans* isomer of retinol, and extracted $K_d$s of 259, 169, and 145 nM for retinol binding to hSAA1, mSAA1, and mSAA3, respectively (*Figure 2B,E*). These values are similar to binding affinities calculated for human serum retinol binding protein (hRBP) (*Cogan et al., 1976*) (*Figure 2—figure supplement 1*). Thus, SAAs bind retinol tightly, with affinities similar to that of a known retinol binding protein.

Retinoic acid lacks intrinsic fluorescence but can quench inherent protein fluorescence due to energy transfer from tryptophan residues (*Cogan et al., 1976*). We therefore measured retinoic acid binding using a modified fluorescence assay that monitored quenching of protein fluorescence

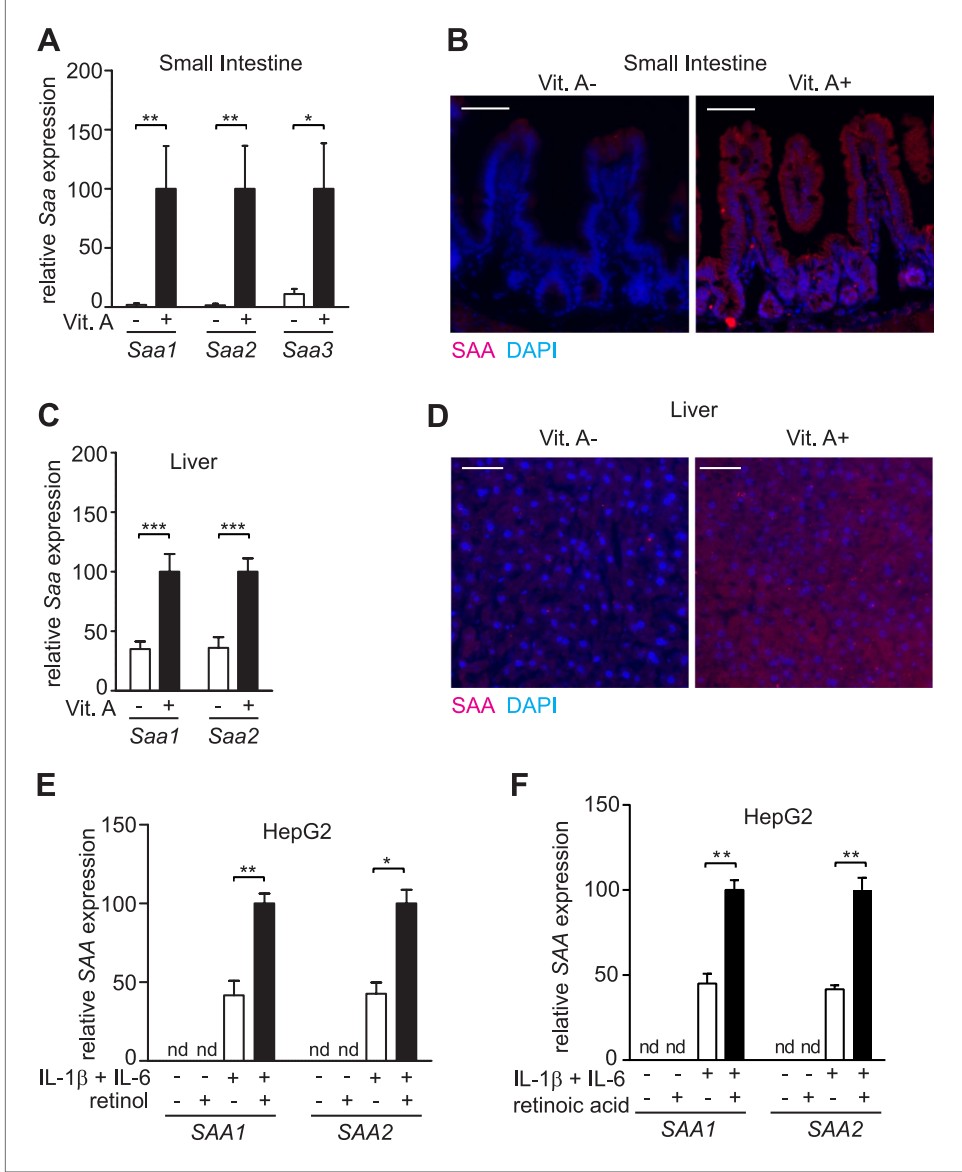

**Figure 1**. SAA expression requires dietary vitamin A. (**A**) Mice were maintained on a normal (Vit. A+) diet or a vitamin A-deficient (Vit. A−) diet as in *Figure 1—figure supplement 1*. Ileal *Saa* expression was quantified by Q-PCR (primer sequences are given in *Table 1*). N = 3–5 mice per condition. (**B**) Ileal sections were stained with anti-SAA antibody ('Materials and methods') and anti-rabbit IgG-Cy3 (red), and counterstained with DAPI (blue). Scale bar = 50 μm. (**C**) Q-PCR determination of *Saa* expression levels in livers of mice on a normal or vitamin A-deficient diet. N = 5 mice/condition. (**D**) Liver sections were stained with anti-SAA antibody and anti-rabbit IgG-Cy3 (red), and counterstained with DAPI (blue). Scale bars = 50 μm. (**E** and **F**) Analysis of SAA expression in HepG2 cells. Cells were cultured in retinoid-free medium and then treated with IL-1β and IL-6 and/or 1 μM retinol (**E**) or 100 nM retinoic acid (**F**). *SAA* expression was determined by Q-PCR. N = 3 independent experiments. Mean ± SEM is plotted. nd, not detected. *p < 0.05; **p < 0.01; ***p < 0.001. p values were determined by two-tailed Student's *t* test.

The following figure supplements are available for figure 1:

**Figure supplement 1**. Intestinal *Saa1* and *Saa2* are differentially regulated by dietary vitamin A.

**Figure supplement 2**. Retinoid supplementation stimulates *Saa* expression in intestine and liver.

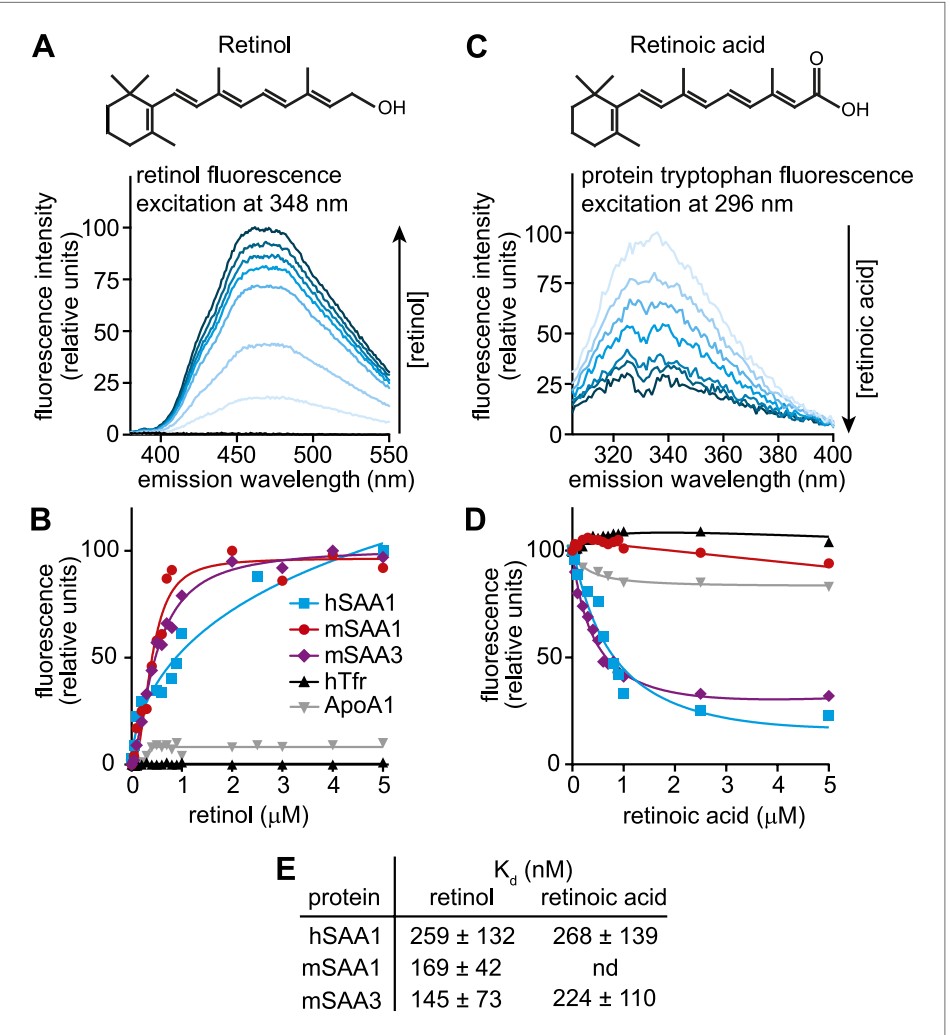

**Figure 2**. Human and mouse SAAs bind retinol. (**A**) Retinol exhibits intrinsic fluorescence that is enhanced upon binding to proteins through energy transfer from tryptophan residues. All-*trans*-retinol was titrated into mSAA3 and fluorescence emission was monitored following excitation at 348 nm. The chemical structure of retinol is shown. (**B**) All-*trans*-retinol was titrated into hSAA1, mSAA1, mSAA3, human transferrin (hTfr; negative control), and apolipoprotein A1 (ApoA1; negative control). Binding was quantified by monitoring retinol fluorescence at 460 nm following excitation at 348 nm as in (**A**). Plots are representative of five independent experiments. (**C**) Retinoic acid lacks intrinsic fluorescence, but can quench intrinsic protein fluorescence due to energy transfer from tryptophan residues (*Cogan et al., 1976*). All-*trans*-retinoic acid was titrated into mSAA3 and fluorescence quenching was monitored following excitation at 296 nm. The chemical structure of retinoic acid is shown. (**D**) All-*trans*-retinoic acid was titrated into hSAA1, mSAA1, mSAA3, hTfr, and ApoA1. Fluorescence emission was monitored at 334 nm with excitation at 296 nm as in (**C**). Plots are representative of three independent experiments. (**E**) $K_d$s were calculated from the binding assay data plotted in (**B**) and (**D**) and were derived from three independent experiments. nd, not determined. Additional ligand binding measurements are provided in *Figure 2—figure supplements 1 and 2*.

The following figure supplements are available for figure 2:

**Figure supplement 1**. Retinol and retinoic acid binding to human retinol binding protein 4 (hRBP4).

**Figure supplement 2**. Additional ligand binding studies on human and mouse SAAs.

(*Figure 2C*). Titration of all-*trans* retinoic acid yielded $K_d$s of 268 and 224 nM for retinoic acid binding to hSAA1 and mSAA3, respectively (*Figure 2D,E*), which are similar to binding affinities calculated for human RBP binding to retinoic acid (*Cogan et al., 1976*) (*Figure 2—figure supplement 1*). There was

weak binding of retinoic acid to mSAA1 and we were unable to calculate a $K_d$ for the interaction (*Figure 2D,E*). Thus, while hSAA1 and mSAA3 bind both retinol and retinoic acid, mSAA1 selectively binds retinol. mSAA1 also showed weak binding to other retinoids, including β-carotene and retinyl acetate, while hSAA1 bound these compounds with $K_d$s of 497 and 347 nM, respectively, and mSAA3 bound β-carotene with a $K_d$ of 159 nM (*Figure 2—figure supplement 2A,B*). All SAA isoforms bound weakly to retinyl palmitate (*Figure 2—figure supplement 2C*). Since long chain retinyl esters (such as retinyl palmitate) are the major form of stored retinoid in the liver (*Vogel et al., 1999*), this suggests that SAAs do not transport retinoids for storage. Although a role for SAAs in cholesterol transport and metabolism has been proposed (*van der Westhuyzen et al., 2005*), we found that cholesterol was unable to competitively inhibit retinol binding to SAAs (*Figure 2—figure supplement 2D*).

To test whether SAAs also associate with retinol in vivo, we sought to purify SAAs from mouse tissues and assay for the presence of associated retinol. SAAs were difficult to purify from the mouse intestine due to the presence of large amounts of contaminating protein, even under conditions where expression of SAAs was maximally induced. However, SAAs constitute a high proportion of serum protein during acute systemic infection (*McAdam and Sipe, 1976*; *Zhang et al., 2005*). We were therefore able to use size exclusion chromatography to recover a SAA-enriched fraction from the sera of mice infected intraperitoneally with *Salmonella typhimurium* for 24 hr (*Figure 3—figure supplement 1A–C*). Mass spectrometry revealed that the SAA-enriched protein fraction was devoid of other known retinol binding proteins (*Figure 3—figure supplement 1D*). Liquid chromatography tandem mass spectrometry (LC-MS/MS) indicated the presence of retinol in the SAA-enriched fraction (*Figure 3*, *Figure 3—figure supplement 2A–C*) in a molar ratio of ~1 mol retinol/4 mol SAA (*Figure 3*, inset). In these analyses, retinoic acid was not detected, and retinol was not detected in the equivalent serum fraction recovered from *Saa1/2$^{-/-}$* mice (*Eckhardt et al., 2010*) (*Figure 3*), suggesting that the retinol was preferentially associated with SAA.

## Mouse SAA3 forms a tetramer with a hydrophobic central channel

SAAs lack sequence homology to the two known families of retinol binding proteins: cellular retinol binding proteins (CRBP) and serum retinol binding proteins (RBP) (*Blaner, 1989*; *Noy, 2000*). Thus, the three-dimensional structures of CRBP and RBP proteins (*Newcomer et al., 1984*; *Cowan et al., 1993*) provide no direct insight into the structural basis for retinol binding by SAAs. To understand how SAAs bind retinol, we therefore determined the three-dimensional structure of recombinant mSAA3 by X-ray crystallography. The protein was crystallized in a $P6_2$ space group with two subunits in the asymmetric unit, and the structure was determined to a resolution of 2 Å by single-wavelength anomalous dispersion (SAD) phasing using a selenomethionyl-derivatived crystal (*Figure 4A*; *Table 2*). The crystal structure reveals that mSAA3 is highly α-helical (*Figure 4A*), as predicted on the basis of its primary sequence (*Figure 4—figure supplement 1*) (*Stevens, 2004*). The structure is very similar to the recently determined structure of human SAA1.1 (*Lu et al., 2014*), an isoform that has a marked tendency to form pathogenic amyloid fibrils (*Yu et al., 2000*). Like the SAA1.1 structure, the mSAA3 structure consists of four α-helices, designated α1-4 from the N- to the C-termini, forming a cone-shaped four-helix bundle with a comparatively longer α1. The helices form two sets of antiparallel helices, α1-α2 and α3-α4, connected by a very short loop (*Figure 4A*). The monomer is stabilized by an extensive network of hydrogen bonding interactions among conserved residues and water molecules in the interior of the monomer. As in the SAA1.1 structure, the C-terminal tail wraps around the helix bundle making a number of hydrogen bonding interactions that add to monomer stability, underscoring the importance of the C-terminal tail.

Size exclusion chromatography and cross-linking experiments showed that mSAA3 forms a tetramer in solution (*Figure 5A,B*). Consistent with these findings, analysis of the mSAA3 crystal structure using the Protein Interfaces, Surfaces and Assemblies (PDBePISA) server (http://www.ebi.ac.uk/msd-srv/prot_int/) yielded a tetrameric quaternary structure (*Figure 4B*; *Table 3*). This is in contrast to the hexameric structure derived for SAA1.1 (*Lu et al., 2014*). There are several potential reasons for the discrepancy in the oligomeric structures. First, it has been suggested that different SAAs can adopt different oligomeric states (*Wang et al., 2002*, *2011*). SAA1.1 was also observed to produce a ~43 kDa species in solution (*Lu et al., 2014*), suggesting that SAA1.1 may in part adopt a tetrameric state. Second, SAA1.1 has a more hydrophobic N-terminus than mSAA3, which is thought to be a determinant of amyloidogenicity (*Yu et al., 2000*; *Lu et al., 2014*). Third, the crystallized SAA1.1 protein retained the hexa-histidine tag (*Lu et al., 2014*), which may have contributed to the difference in oligomeric state.

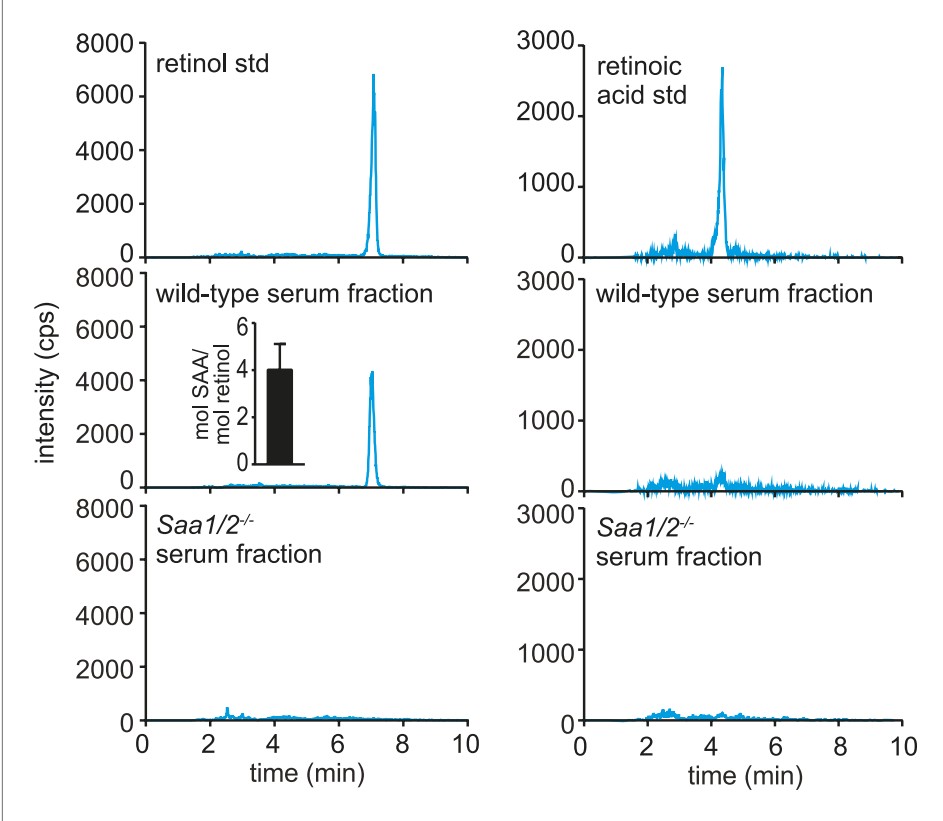

**Figure 3**. Serum SAA is associated with retinol in vivo. Wild-type or *Saa1/2*[−/−] mice were infected intraperitoneally with *S. typhimurium* and serum was collected 24 hr later. The serum was fractionated by size exclusion chromatography and a major SAA-containing fraction from wild-type mice was identified by Western blot (***Figure 3—figure supplement 1***). The SAA-containing fraction and the equivalent serum fraction from *Saa1/2*[−/−] mice were hexane-extracted and analyzed by LC-MS/MS against retinol and retinoic acid standards. Additional support for the identification of retinol is provided in ***Figure 3—figure supplement 2***. The LC-MS/MS chromatograms of daughter ion 93 are shown. mol SAA/mol retinol is shown in the inset. Data are representative of duplicate experiments with triplicate samples in each experiment.
The following figure supplements are available for figure 3:

**Figure supplement 1**. Size-exclusion chromatography and mass spectrometry analysis of SAA-containing serum fractions.

**Figure supplement 2**. Mouse SAA is associated with retinol in the serum following infection.

The mSAA3 tetramer is formed by two sets of tightly associated dimers (***Figure 4B,C***). The dimers pack against each other through an aromatic interface formed by W71, as well as a non-polar interaction involving V75 residues (***Figure 4C,E***). The dimer is formed by two identical chains oriented pseudo-anti-parallel to each other, and is held together by two pairs of tight hydrogen bond interactions between K74 and D78 residues on oppositely oriented α3 helices (***Figure 4C,D***). The interaction is also supported by a set of aromatic interactions between F99 and W103 of the α4 helices (***Figure 4C,D***). These interactions result in tightly held α3 helices composed primarily of non-polar residues, thus forming the inside hollow pocket of the tetramer. The hollow pocket is surrounded by the remaining helices, creating a hydrophobic interior that is protected from the external aqueous environment.

## The mouse SAA3 central channel is predicted to accommodate retinol
Retinol is a highly apolar lipid-like molecule consisting of a β-ionone ring, an isoprenoid tail, and a hydroxyl group. Thus, it requires a non-polar environment for transport among tissues and within cells.

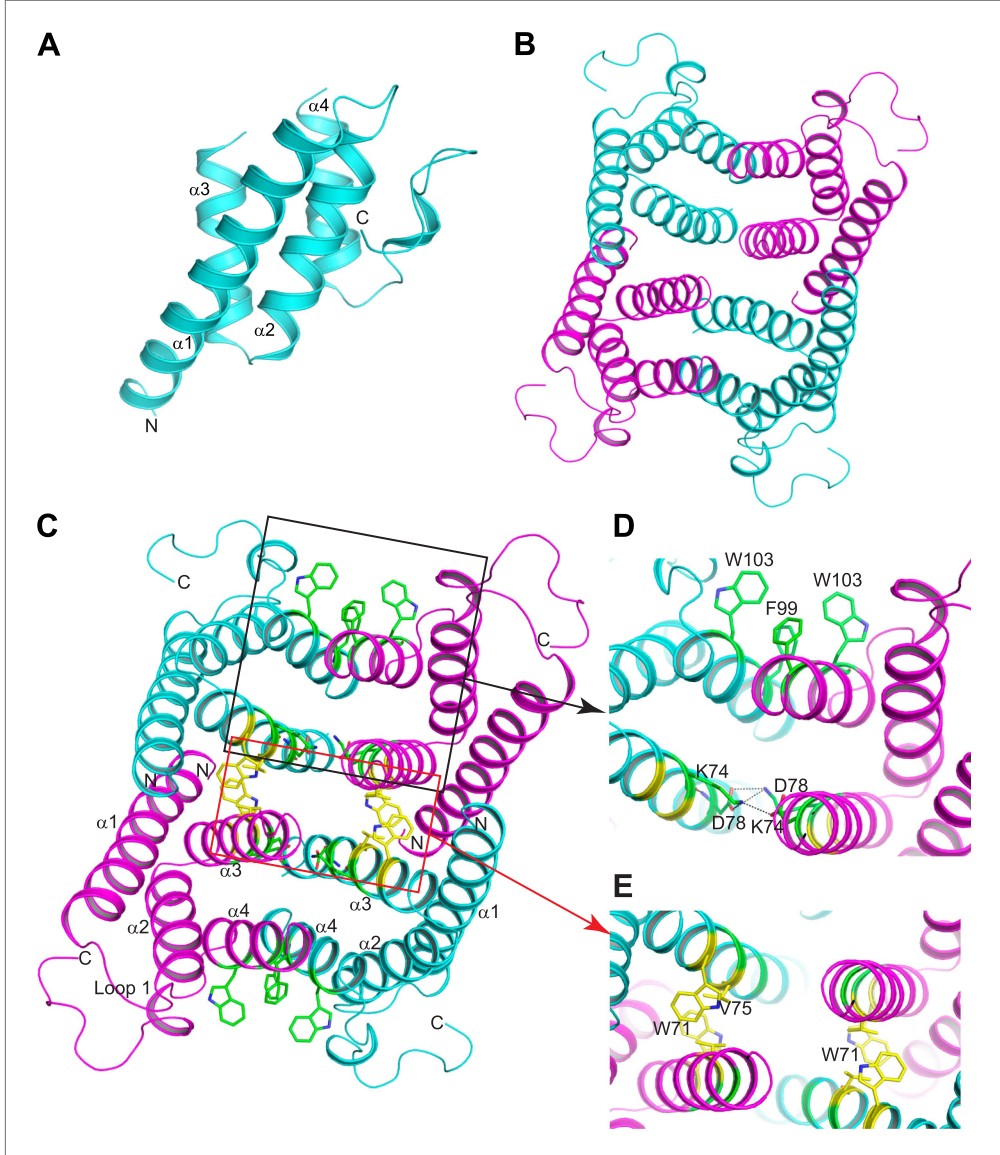

**Figure 4**. Structure of mSAA3 and molecular contacts within the tetrameric unit. (**A**) Structure of the mSAA3 monomer (side view), with helices and termini labeled. (**B**) Top view of the tetrameric mSAA3 structure. Chains forming dimer pairs are colored cyan and magenta. In (**C**), helices α1-4 and the N- and C-termini of two monomers are labeled. Residues that make dimer contacts are shown as green sticks while residues involved in tetramer stabilization are shown as yellow sticks. (**D** and **E**) Magnified regions of a dimer interface (**D**) and tetramer interface (**E**) are shown. Views are slightly rotated so that the interactions can be clearly visualized. Crystal structure data collection and refinement statistics are provided in *Table 2*; alignments of mouse and human SAAs are shown in *Figure 4—figure supplement 1*; parameters from the protein Interfaces, Surfaces, and Assemblies (PISA) analysis showing a tetrameric state are provided in *Table 3*.

The following figure supplement is available for figure 4:

**Figure supplement 1**. Sequence alignment of mouse and human SAAs.

Structures of known retinol binding proteins, including serum RBP, indicate that these proteins consist mainly of β-sheet secondary structures forming a β-barrel tertiary structure, with the retinol molecule held in an interior non-polar binding pocket (*Newcomer et al., 1984*). In contrast, mSAA3 is α-helical and oligomerizes to form a hollow, largely non-polar interior that could serve as a binding pocket for a non-polar small molecule (*Figure 6A*). We were unable to obtain mSAA3 crystals with the bound

**Table 2.** Crystal structure data collection and refinement statistics

| Data collection | |
|---|---|
| Space group | P6$_2$ |
| Cell dimensions (Å) | a = b = 78.33, c = 62.32 |
| | α = β = 90°, γ = 120° |
| Wavelength (Å) | 0.9794 |
| R$_{sym}$ or R$_{merge}$ (%) | 8.4 |
| Resolution (Å)* | 50–2.05 (2.09–2.05) |
| I/σI | 19.19 (3.23) |
| Completeness (%) | 99.8 (97.3) |
| Redundancy | 6.2 (5.4) |
| Refinement | |
| No. reflections | 12,206 |
| Resolution (Å)* | 39.17–2.06 (2.14–2.06) |
| R$_{work}$/R$_{free}$ | 0.17/0.21 (0.16/0.19) |
| No. atoms | |
| Protein | 1608 |
| Ligand/ion | 3 |
| Water | 61 |
| R.m.s. deviations | |
| Bond lengths (Å) | 0.0077 |
| Bond angles (°) | 0.932 |

*Highest resolution shell is shown in parenthesis.

retinol ligand as retinol is highly unstable (*Barua and Furr, 1998*) and the crystals required several weeks to grow. However, a ligand docking analysis using SwissDock (*Grosdidier et al., 2011*) indicated that retinol can be docked in this hydrophobic pocket with favorable free energy (~ −7 kcal/mol) and FullFitness values (*Zoete et al., 2010*) (~ −2400 kcal/mol) (*Figure 6B–D*). Consistent with this prediction, introducing a Trp71Ala (W71A) mutation in the mSAA3 hydrophobic core reduced the affinity of mSAA3 for retinol (*Figure 6E*). Thus, the mSAA3 structure supports our biochemical data showing a retinol binding function for SAAs and explains how mSAA3 could bind retinol.

## Discussion

Numerous biochemical, physiologic, and epidemiologic studies have shown that vitamin A and its derivative retinol are essential for the development of robust immunity (*Stephensen, 2001*). Retinol and retinoic acid are critical for the development of innate and adaptive immunity (*Lawson and Berliner, 1999*; *Mucida et al., 2007*; *Hall et al., 2011*; *Spencer et al., 2014*), and also promote maintenance and repair of epithelial barriers (*Osanai et al., 2007*). However, a prominent response to acute infection is the marked decline in serum RBP (*Rosales et al., 1996*), which paradoxically occurs at a time of increased demand for retinol to support development of immunity and barrier defense. Thus, it has been unclear how retinol is transported among tissues following an acute microbial challenge.

We propose that SAAs fulfill this role, supported by several lines of evidence. First, SAAs are strongly induced by microbial exposure at sites of retinol uptake (intestine) and storage (liver), and are present at high levels in the circulation following microbial challenge (*Chiba et al., 2009*; *Ivanov et al., 2009*; *Reigstad et al., 2009*). Second, we have shown that SAAs bind retinol at nanomolar affinity in vitro, and that serum SAAs circulate in association with retinol. Third, the three dimensional structure of mSAA3 exhibits a hollow hydrophobic binding pocket, providing structural insight into how SAAs bind retinol.

Although the precise tissue targets of circulating retinol-bound SAAs remain under investigation, several observations support the idea that SAAs promote immunity to infection. First, *Saa1/2*$^{-/-}$ mice exhibit increased susceptibility to chemically-induced colitis in mice (*Eckhardt et al., 2010*), suggesting that SAAs contribute to intestinal immunity. Second, studies in zebrafish show that commensal microbiota stimulate neutrophil migration through induction of SAA (*Kanther et al., 2013*). Third, we found that intraperitoneal infection of *Saa1/2*$^{-/-}$ mice with *S. typhimurium* resulted in higher bacterial loads in liver and spleen as compared to wild-type mice (*Figure 7A,B*), suggesting that SAAs also contribute to systemic immunity.

SAA4 is an SAA isoform that is expressed in the livers of healthy, non-infected mice and humans (*de Beer et al., 1991*, *1994*, *1995*). SAA4 circulates at concentrations that are markedly lower than those observed for SAA1 and 2 following acute infection (*de Beer et al., 1995*) but are similar to the concentrations of RBP in uninfected individuals (*Willett et al., 1985*; *Friedman et al., 1986*). SAA4 is 54–56% homologous to SAA1, 2, and 3, and retains the hydrophobic amino acids that are predicted to line the hydrophobic binding pocket in SAA3. Further, homology modeling using the mouse SAA3 structure and the mouse SAA4 sequences yields a SAA4 model with a highly similar predicted structure (alignment score of 0.1 and a Global Model Quality Estimate of 0.73). Thus, the possibility that

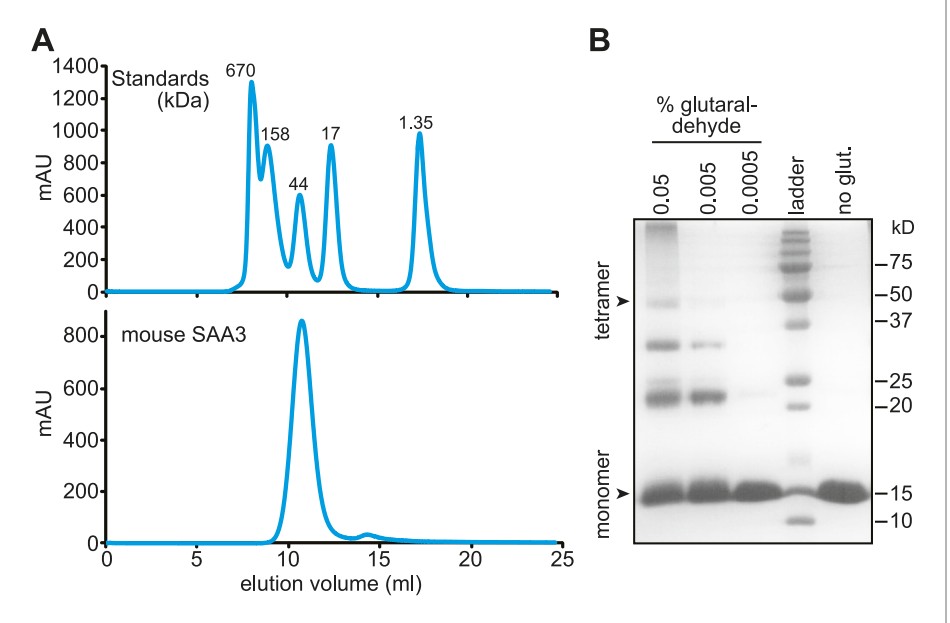

**Figure 5**. Mouse SAA3 is tetrameric in solution. (**A**) Size exclusion chromatography profile of purified mouse SAA3 on a Superdex 75 10/300 GL column. Elution of standards (BioRad) is shown in the top panel and elution of SAA3 is shown in the bottom panel. mSAA3 elutes at a position consistent with a tetramer (monomer is 12.2 kDa). (**B**) Cross-linking analysis of mSAA3. Purified mSAA3 was cross-linked with glutaraldehyde and analyzed by SDS-PAGE. DOI: 10.7554/eLife.03206.016

SAA4 is a retinol binding protein that functions to transport retinol in healthy, non-infected animals will be a subject for future investigation.

SAA1, 2, and 3 are markedly induced in the intestinal epithelium by the microbiota (*Ivanov et al., 2009*; *Eckhardt et al., 2010*; *Reigstad and Bäckhed, 2010*), and thus our findings may provide insight into how the intestinal microbiota regulates host immunity and inflammation. Previous studies have suggested that SAAs promote Th17 cell development in response to specific components of the microbiota, such as segmented filamentous bacteria (*Ivanov et al., 2009*). Consistent with a function for SAAs in retinol binding and transport, retinol/retinoic acid is required to elicit Th17 cell responses to infection and mucosal vaccination (*Hall et al., 2011*). A key question is whether there are tissue-specific effects of intestinal epithelial SAAs, or whether the intestinal SAAs enter the circulation with bound retinol acquired directly from the diet. For example, intestinal epithelial SAAs could be involved in the direct delivery of retinol from epithelial cells to underlying immune cells in the lamina propria, or from epithelial cells to mucosal lymphoid tissues.

Altogether, our results provide insight into the biological function of SAAs, reveal a new protein architecture that supports retinol binding, and suggest how retinol is transported among cells and tissues during infection. These findings may prove useful in designing new strategies for enhancing resistance to infection and/or controlling inflammation during disease.

## Materials and methods

### Animals

C57BL/6 wild-type mice were maintained in the barrier at the University of Texas Southwestern Medical Center. *Saa1/2⁻/⁻* mice were obtained from Dr Frederick C de Beer (*Eckhardt et al., 2010*) at

**Table 3.** Parameters from the Protein Interfaces, Surfaces, and Assemblies (PISA) analysis

| Parameter | Value |
| --- | --- |
| Multimeric state | 4 |
| Composition | $A_2B_2$ |
| Dissociation pattern | 2(AB) |
| Surface area, Å$^2$ | 19485.8 |
| Buried area, Å$^2$ | 6125.7 |
| $\Delta G_{intrinsic}$, kcal/mol | −79.8 |
| $\Delta G_{diss}$, kcal/mol | 11.1 |
| T$\Delta S_{diss}$, kcal/mol | 12.6 |

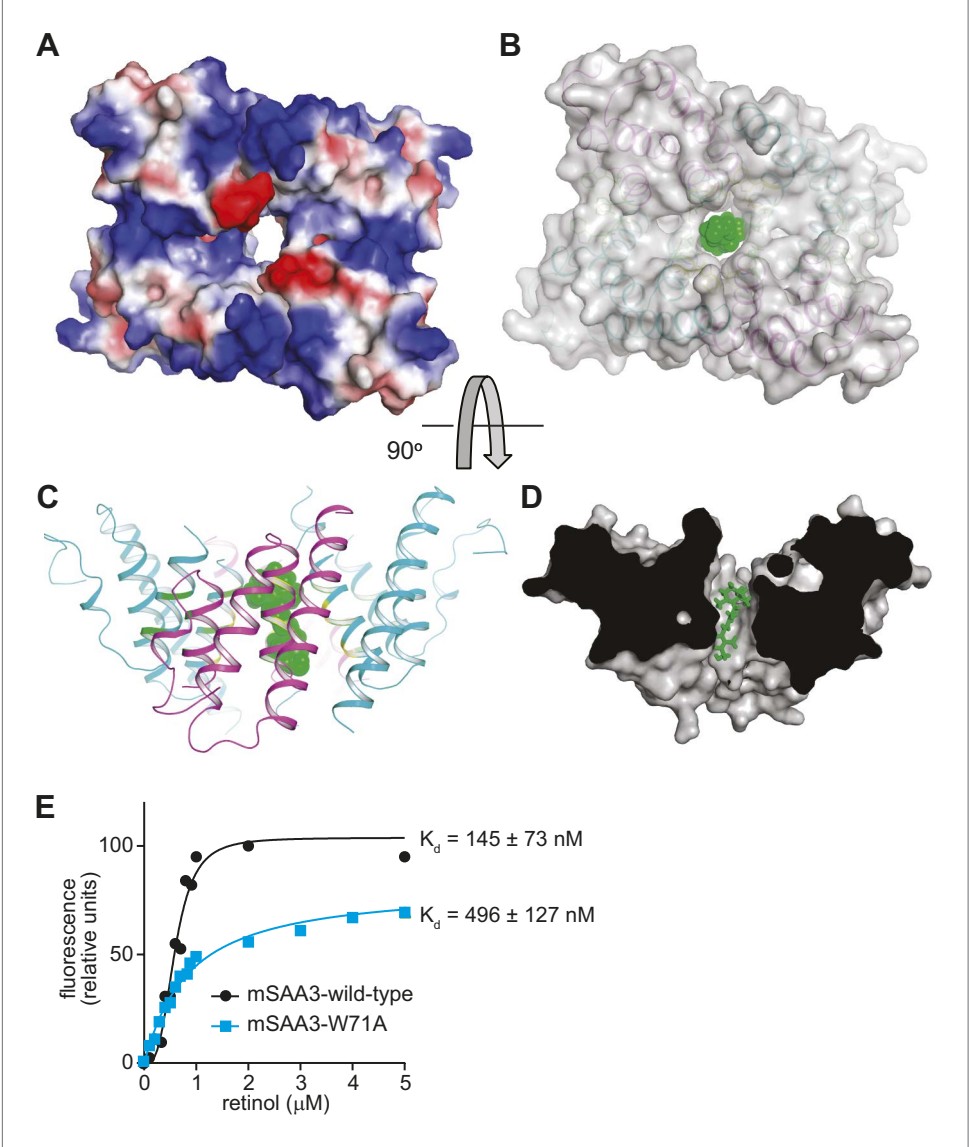

**Figure 6**. The mSAA3 tetramer forms a hollow hydrophobic binding pocket that can accommodate retinol. (**A**) A surface rendering of the tetramer showing the interior cavity, with the electrostatic potential displayed using a color gradient ranging from negative (red) to neutral (white) to positive (blue). The orientation is similar to that in *Figure 4B*. (**B**–**D**) Different views of a retinol molecule docked in the putative ligand-binding pocket. (**B**) A semi-transparent surface representation of the protein in the same orientation as *Figure 4B*, with a cartoon trace. Retinol atoms are represented as green spheres. The views in (**C**) and (**D**) are rotated by approximately 90° in the horizontal plane relative to (**A**) and (**B**), and (**B**) is rotated by approximately 90° in the vertical plane relative to (**C**). In (**D**), a surface model of the protein is shown, sliced close to the binding pocket. Retinol atoms are shown as sticks. (**E**) Wild-type or Trp71Ala (W71A) mutant mSAA3 was assayed for retinol binding as described in *Figure 2*. Representative plots and $K_d$s were calculated from the binding assay data and were derived from five independent experiments.

the University of Kentucky and were maintained in the barrier at the University of Texas Southwestern Medical Center. 6–12 weeks old mice were used for all experiments. Experiments were performed using protocols approved by the Institutional Animal Care and Use Committees of the UT Southwestern Medical Center.

## Antibodies and reagents

Recombinant mSAA1 and 3 were expressed and purified as described below. hSAA1 protein was from PeproTech (Rocky Hill, NJ) and resuspended as recommended. The protein is a consensus

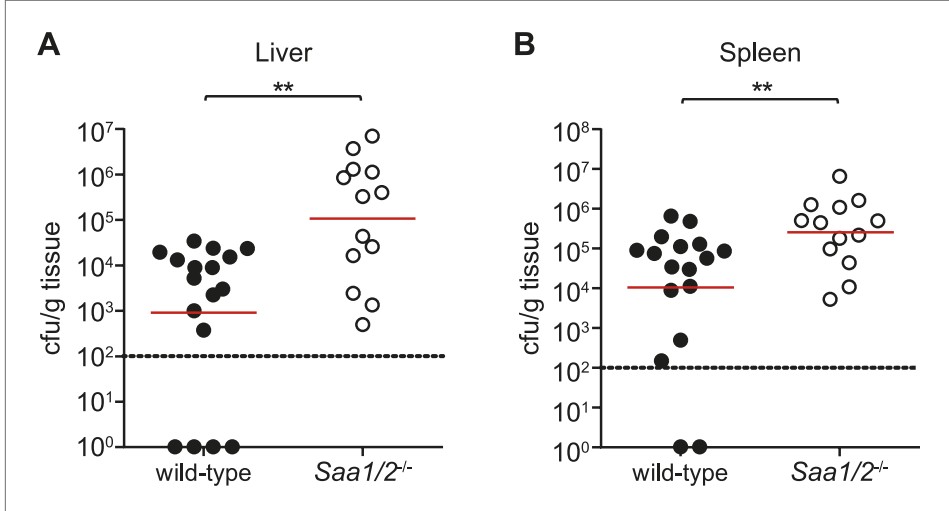

**Figure 7**. *Saa1/2−/−* mice have higher bacterial burdens following *S. typhimurium* infection. 10 week old wild-type and *Saa1/2−/−* mice were inoculated intraperitoneally with 10,000 cfu of S. typhimurium. Livers (**A**) and spleens (**B**) were collected after 24 hr and analyzed for bacterial counts by dilution plating. Combined results from two independent experiments are shown. Each point represents one mouse and geometric means are indicated. Dotted line indicates limit of detection. \*\*p < 0.01 using the Mann–Whitney test.

SAA molecule corresponding to human apo-SAA1α except for the presence of an N-terminal methionine and substitution of asparagine for aspartic acid at position 60 and arginine for histidine at position 71. Anti-SAA antiserum was raised against purified recombinant mSAA1. Retinol, retinoic acid, β-carotene, retinyl acetate, retinyl palmitate, and cholesterol were from Sigma-Aldrich (St. Louis, MO) and were reconstituted into ethanol, DMSO, or dioxane, depending on the experiment. IL-1β and IL-6 were from Invitrogen.

## Vitamin A depletion
Vitamin A-deficient (TD.09838) and control (~20,000 IU vitamin A/kg; TD.09839) diets were purchased from Harlan Laboratories (South Easton, MA). At day 10 of gestation, pregnant females were placed on the standard diet or the vitamin A-deficient diet (*Hall et al., 2011*). Mothers and pups were maintained on the diets until weaning, and pups stayed on the diet for two additional months prior to sacrifice.

## In vivo retinoic acid reconstitution
The protocol was adapted from a previously described procedure (*Hall et al., 2011*). A total of 250 µg of all-*trans*-retinoic acid (Sigma-Aldrich) was resuspended in 30 µl of biotechnology performance certified DMSO (Sigma-Aldrich). The suspension was administered daily by intraperitoneal injection to vitamin A-deficient mice over the course of 3 days. 24 hr following the third injection, mice were sacrificed and tissues were harvested. Control mice received an injection of the DMSO vehicle.

## Intestinal explant culture
Terminal ileum (5 cm) was collected from mice post-sacrifice and flushed with a solution of phosphate-buffered saline with penicillin (100 units/ml) and streptomycin (100 µg/ml). Ileal segments were cultured on equilibrated cell culture plate inserts (PIHA03050; Millipore) at 37°C and 95% oxygen for 6 hr in Dulbecco's modified Eagle's medium (4 g/l glucose and L-glutamine; Invitrogen, Carlsbad, CA) supplemented with 10% charcoal-stripped heat-inactivated fetal bovine serum (Gibco, Carlsbad, CA), 10% NCTC135 media (Sigma), 25 mM HEPES, 100 units/ml penicillin, 100 µg/ml streptomycin, and either 0.1% DMSO or 1 µM retinol in 0.1% DMSO. After culture for 6 hr, segments were flash-frozen and processed for total RNA extraction.

## Microarray experiments
Total RNAs were isolated from mouse ileum using the Qiagen Midi-Prep RNA isolation kit. For each condition, RNA was isolated from two independent groups of five to eight mice. The RNAs in each

group were pooled and used to generate biotinylated probes for microarray analysis. Probes were hybridized to Affymetrix Mouse Genome 430 2.0 GeneChips in the University of Texas Southwestern Microarray Core.

To identify genes that are differentially expressed between germ-free and conventional mice, we performed two-way comparisons between germ-free and conventional groups, with germ-free samples designated as baseline. Raw data were imported into Affymetrix (Santa Clara, CA) GeneChip software for analysis, and previously established criteria were used to identify differentially expressed genes (*Cash et al., 2006*). Briefly, a twofold difference was considered significant if three criteria were met: (1) the GeneChip software returned a difference call of increased or decreased; (2) the mRNA was called present by GeneChip software in either germ-free or conventional samples; and (3) the difference was observed in duplicate microarray experiments. We performed a similar analysis to identify genes that are differentially regulated between mice fed a normal diet vs those fed a vitamin A-deficient diet. Finally, we identified 19 genes that were differentially regulated by colonization status and by dietary vitamin A content. Signal intensity data for this group of 19 genes were converted to Z-scores ($z = (x − \mu)/\sigma$, where x = signal intensity, $\mu$ = mean signal intensity for all samples, and $\sigma$ = SD across all samples), which were visualized as heatmaps using Java TreeView software.

## Quantitative PCR

Total RNA was isolated from homogenized tissues or cells using the Qiagen RNeasy RNA isolation kit. Random primed cDNAs were assayed by SYBR Green-based real-time PCR using *SAA*-specific primers as given in *Table 1*. Signals were normalized to 18S rRNA or *Gapdh*.

## Immunofluorescence analysis

Zinc-fixed, paraffin embedded tissue sections were stained with anti-SAA antiserum raised against purified recombinant mSAA1 and detected using a goat anti-rabbit IgG Cy3 conjugate (Biomeda). Tissues were counterstained with DAPI and images were captured on a Zeiss AxioImager M1 Microscope.

## Cell culture

HepG2 cells were purchased from ATCC. Cells were maintained in 1X DMEM, 10% FBS (or charcoal stripped FBS), 1X Penstrep, 1X glutamax, and 1X sodium pyruvate. Cells were maintained at 5% $CO_2$. Prior to addition of retinoids, the cells were grown overnight in DMEM containing 10% charcoal-stripped FBS (to removes retinoids) and were treated with 1 µM retinol or 100 nM retinoic acid, 10 ng/ml of IL-1β, and 10 ng/ml IL-6.

## Expression and purification of recombinant SAAs

Genes encoding mouse SAA1 and SAA3 (minus the signal sequence) were cloned into the pET28(a)+ expression vector between *NdeI* and *BamHI* restriction endonuclease sites, with an N-terminal hexa-histidine tag followed by a thrombin cleavage site and a C-terminal stop codon. Proteins were expressed in *Escherichia coli* BL21-CodonPlus (DE3)-RILP cells (Stratagene, La Jolla, CA) by induction with 0.4 mM isopropyl-β-D-galactoside (IPTG) for ~3 hr at 25°C for mSAA1, and at 37°C for mSAA3. Cells were harvested by centrifugation at 4500×g for 25 min at 4°C and re-suspended in lysis buffer (50 mM $NaH_2PO_4$, 500 mM NaCl, 10 mM imidizole for mSAA1 and 500 mM NaCl, 50 mM Tris pH 8.0, 10 mM imidazole, 15 mM β-mercaptoethanol for mSAA3). After sonication, decyl maltopyranoside (DM) (Avanti Polar Lipids, Alabaster, AL) was added to a final concentration of 40 mM and incubated for ~3 hr at 4°C. The mixture was pelleted by centrifugation at 10,000×g for 30 min and the supernatant loaded onto a $Ni^{2+}$ metal affinity column (Qiagen, Valencia, CA) pre-equilibriated with 4 mM DM in lysis buffer. Non-specific contaminants were washed away with 25 mM imidazole in DM buffer and the protein was eluted in DM buffer containing 300 mM imidazole. All subsequent buffers do not include detergent in order to completely remove detergent. The eluate was desalted with a HiTrap desalting column (GE Life Sciences, Pittsburgh, PA) into a 200 mM NaCl buffer. Thrombin (Roche, Basel, Switzerland) was then added (~1 unit/1.2 mg protein) and incubated overnight at 4°C. Undigested protein was removed by passing the overnight digest over $Ni^{2+}$ affinity matrix, collecting only the flow through. The eluate was concentrated in a 3 K cutoff Amicon Ultra centrifugal device (Millipore, Billerica, MA) and further purified by size exclusion chromatography on either a HiLoad Superdex 200 or a Superdex 75 (10/30) column (GE Life Sciences, Pittsburgh, PA), in 100 mM NaCl, 20 mM Tris pH 8, 15 mM a β-mercaptoethanol and 5% glycerol. For crystallization, mSAA3 was concentrated to ~3 mg/ml.

## Mutagenesis

The mSAA3-W71A mutant was generated using the QuickChange II site-directed mutagenesis kit (Agilent Technologies, Santa Clara, CA). Protein expression, purification, and retinol binding assays were done as for wild-type mSAA3.

## Binding assays

Steady state fluorescence was measured using a QuantaMaster 40 spectrofluorometer (Photon Technology International, Edison, NJ) and FelixGX software program. For retinol titrations, samples were excited at 348 nm and emissions monitored at 460 nm. For retinoic acid and other retinoids, samples were excited at 296 nm and tryptophan quenching was monitored by emission at 334 nm. Experiments on mSAA1 and mSAA3 were done in 25 mM Tris pH 8.0, 100 mM NaCl, and 4 mM decyl maltopyranoside (DM). Experiments with hSAA1, human ApoA1, and human transferrin were conducted in PBS. For cholesterol competition assays, saturating concentrations of retinol were added to hSAA1, mSAA1, and mSAA3, and fluorescence quenching was monitored by emission at 334 nm. 10 μM cholesterol was added to the assay and inhibition of fluorescence quenching by retinol was monitored. All assays were done using a protein concentration of 0.5 μM.

## S. typhimurium infections

*Salmonella enterica* Serovar Typhimurium (SL1344) was grown overnight in Luria Broth at 37°C. Mice were infected intraperitoneally with $1 \times 10^4$ organisms per mouse. Mice were sacrificed after 24 hr and tissues and serum were collected for experiments.

## Size exclusion chromatography of serum SAA

Serum was pooled from 3–5 mice and 500 μl was separated by size-exclusion chromatography on a Superdex 75 HiLoad 16/60 column (GE Life Sciences). Peak fractions were analyzed by SDS-PAGE and stained with Coomassie Blue. Duplicate samples were analyzed by Western blot with anti-SAA antibody to identify peak fractions containing SAA protein.

## Mass spectrometry analysis of retinol and retinoic acid

Retinoid extraction was modified and scaled from a previously described procedure (*McClean et al., 1982*). SAA-containing fractions purified by size exclusion chromatography were pooled, added to an equal volume of 1:1 1-butanol:acetonitrile, and vortexed for 60 s. 20 μl of 20.6 M $K_2HPO_4$ was added for each 1 ml of pooled fractions. Samples were then vortexed 30 s and 5 ml of hexane per 1 ml sample was added. Samples were vortexed for another 30 s and centrifuged at 1,000×*g* for 5 min and the top organic phase was dried in a nitrogen evaporator (Organomation Associates, Berlin, MA). Samples were prepared the day before the assay and stored at 80°C. Standard solutions were resuspended in ethanol and prepared fresh for every use. Standard curves were generated by spiking retinol or retinoic acid into 1 ml of 20 mM Tris pH 8.0, 100 mM NaCl and processed as for serum samples. Samples were resuspended in 200 μl of acetonitrile before injection. Compound levels were monitored by LC-MS/MS on an AB/Sciex (Framingham, MA) 4000 Qtrap mass spectrometer coupled to a Shimadzu (Columbia, MD) Prominence LC after a 20 μl injection. The compounds were detected using electrospray ionization (ESI) with the mass spectrometer in MRM (multiple reaction monitoring) mode by following the precursor to fragment ion transition 269.2 → 93.1 and 269.2 → 119 for retinol (pos. mode; $[M-H_2O]^+$) and 301.2 → 123.1 for retinoic acid (pos. mode; $M+H^+$). An Agilent (Santa Clara, CA) Eclipse XDB C18 column (150 × 4.6 mm, 5 micron packing) was used for chromatography with the following conditions: mobile phase A: acetonitrile:methanol:$H_2O$:formic acid (55:33:12:.01); mobile phase B: acetonitrile:formic acid (100:0.01). Over a total run time of 18 min, the following gradient was applied: 0 to 3 min 50% B; 3 to 10 min gradient to 100% B; 10 to 17 min 100% B; 17 to 18 min gradient to 50% B. Stoichiometries of the SAA-retinol association were determined by quantifying serum retinol and serum SAA. Total serum retinol was calculated based on peak areas from the mass spectrometer analysis in samples compared to a retinol standard curve. Serum SAA was quantified by Western blot analysis with anti-SAA antiserum and densitometry.

## Determination of the mSAA3 crystal structure

Crystals were grown by sitting-drop vapor diffusion at 20°C by mixing equal volumes of protein and reservoir. An initial hit was obtained in 30–40% 2-methyl-2,4-pentanediol (MPD), 0.1 M sodium acetate pH 4.5 after more than a month. Further refinement yielded better crystals at 75–80% MPD. Crystals

were directly flash frozen, the MPD serving as a cryoprotectant. Crystals were of space group P6$_2$ with cell dimensions $a$ = 78.327 Å, $c$ = 62.319 Å and two subunit copies in the asymmetric unit. Selenomethionyl-derivatived crystals were grown the same way as the native protein crystals, except the culture media used was defined media with selenomethionine additive (Molecular Dimensions, Altamonte Springs, FL) and purification buffers after the last Ni$^{2+}$ column elution contained 10 mM DTT (instead of β-mercaptoethanol) and 0.5 mM EDTA.

Data were collected at 100 K under a nitrogen gas stream at the Advanced Photon Source (APS) beamlines 19ID or 23IDD of the Argonne National Laboratory. Single-wavelength anomalous dispersion (SAD) data were collected at the selenium K edge (0.9793 Å). Diffraction data were processed with the HKL2000/3000 package (*Otwinowski and Minor, 2013*). Heavy atom substructure as well as initial phases were obtained using the SAD pipeline in the PHENIX crystallographic software package (*Afonine et al., 2012*). This was followed by manual model building in Coot (*Emsley et al., 2010*) interspersed with iterative rounds of rigid body, simulated annealing and individual isotropic B-factor refinement and finally TLS refinement in PHENIX. The structure was determined to a resolution of 2 Å. Data collection and refinement statistics are summarized in *Table 2*.

### mSAA3 cross-linking analysis

Glutaraldehyde (Sigma) was added to varying final concentrations (0.05%, 0.005%, and 0.0005% wt/vol) to purified mSAA3 (0.5 mg/ml in PBS). The reaction mixtures were incubated for ~30 min on ice, quenched with 0.1 M Tris pH 8, and analyzed by SDS-PAGE.

### Statistics

Statistical differences were calculated by the unpaired two-tailed Student's *t* test or Mann–Whitney test using GraphPad Prism software. Results are expressed as the mean ± standard error of the mean (SEM).

## Acknowledgements

We thank Cassie Behrendt Boyd, Charmaine Clements, and Tess Leal for assistance with mouse experiments, the staff at 19-ID and 23-IDD of the Advanced Photon Source (APS) for beamline access, Jun Liao, Youxing Jiang and Nam Nguyen for beamline access and discussions, the UT Southwestern Structural Biology Laboratory core for help with data collection and Diana Tomchick for advice on crystallography. This work was supported by NIH R01 DK070855 (LVH), the Welch Foundation (I-1762 to LVH), a Burroughs Wellcome Foundation Investigators in the Pathogenesis of Infectious Diseases Award (LVH), and the Howard Hughes Medical Institute (LVH). MGD was supported by a UNCF/Merck Postdoctoral Fellowship and a Burroughs Wellcome Fund Postdoctoral Enrichment Program Award, CMZ was supported by NIH Grant T32 AI005284, and SV was supported by Crohn's and Colitis Foundation of America Fellowship Award. Results shown in this report are derived from work performed at Argonne National Laboratory, Structural Biology Center at the APS. GM/CA at the APS has been funded in whole or in part with Federal funds from the National Cancer Institute (Y1-CO-1020) and the National Institute of General Medical Sciences (Y1-GM-1104). Argonne is operated by UChicago Argonne, LLC, for the U.S. Department of Energy, Office of Biological and Environmental Research under contract DE-AC02-06CH11357. Coordinates of the crystallographic structure of mSAA3 have been deposited in the PDB with accession code 4Q5G.

## Additional information

### Funding

| Funder | Grant reference number | Author |
| --- | --- | --- |
| Welch Foundation | I-1762 | Lora V Hooper |
| Howard Hughes Medical Institute | | Lora V Hooper |
| National Institutes of Health | R01 DK070855 | Lora V Hooper |
| Burroughs Wellcome Fund | Minority Enrichment Program | Mehabaw G Derebe |

| Funder | Grant reference number | Author |
| --- | --- | --- |
| UNCF/Merck Postdoctoral Fellowship | | Mehabaw G Derebe |
| Crohn's and Colitis Foundation of America | Career Development Award | Shipra Vaishnava |
| National Institutes of Health | T32 AI005284 | Clare M Zlatkov |
| Burroughs Wellcome Fund | Investigators in the Pathogenesis of Infectious Diseases | Lora V Hooper |

The funders had no role in study design, data collection and interpretation, or the decision to submit the work for publication.

## Author contributions

MGD, CMZ, Conception and design, Acquisition of data, Analysis and interpretation of data, Drafting or revising the article; SG, Acquisition of data, Analysis and interpretation of data, Drafting or revising the article; KAR, SV, GED, JBMM, NSW, Acquisition of data, Analysis and interpretation of data; LVH, Conception and design, Analysis and interpretation of data, Drafting or revising the article

## Ethics

Animal experimentation: Animal subjects research approved by all animal experiments were approved by the Institutional Animal Care and Research Advisory Committee at the University of Texas Southwestern Medical Center, and the approved animal protocol number is 2011-0197. The institutional guidelines for the care and use of laboratory animals were followed.

## Additional files

### Major dataset

The following dataset was generated:

| Author(s) | Year | Dataset title | Dataset ID and/or URL | Database, license, and accessibility information |
| --- | --- | --- | --- | --- |
| Derebe MG, Hooper LV | 2014 | Crystal Structure of mouse Serum Amyloid A3 | 4Q5G; http://www.rcsb.org/pdb/search/structidSearch.do?structureId=4Q5G | Publicly available at the RCSB Protein Data Bank (http://www.rcsb.org/pdb/home/home.do). |

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
