## [Decision Letter]

Thank you for sending your work entitled "Serum amyloid A is a retinol binding protein that transports retinol during bacterial infection" for consideration at *eLife*. Your article has been favorably evaluated by Richard Losick (Senior editor) and 2 reviewers, one of whom is a member of our Board of Reviewing Editors, and one of whom, Andrew MacPherson, has agreed to reveal his identity.

The Reviewing editor and the other reviewer discussed their comments before we reached this decision, and the Reviewing editor has assembled the following comments to help you prepare a revised submission.

Derebe et al describe the induction of the family of serum amyloid A proteins by vitamin A and their role in the transport of retinol and its derivatives. This is a novel and interesting study that has important implications for understanding of the multiple functions of vitamin A in infection and immunity. The work has been carefully performed to a very high standard work and is suitable for publication essentially in its present form.

The starting point of the paper was an Affy transcript abundance analysis that captured the coincidence of ileal gene expression alterations comparing germ-free and colonised mice with alterations depending on whether colonised mice were maintained on a vitamin A-deficient or -sufficient diet. The genes that show consistent expression differences include saa 1 and 2 (please note that I assume that saa 3 should be substituted for one of the saa1 labels, which appear twice in Figure 1—figure supplement 1).

The interpretation that SAA (1 and 3) are transport proteins for retinol is justified by direct binding studies, by elution from serum using the authentic retinol standard to calibrate LC MS and the use of SAA (1 and 2) deficient mice in a strain-combination setup. There is also a neat demonstration of a non-polar binding pocket in crystallographic analysis of SAA3, which is shown directly to form a tetramer through crosslinking studies.

The binding and structural studies are strong and represent the major advance of the paper. The link between vit A and SAA expression provides some valuable functional insight but this part is not as well-developed. The final experiment showing that SAA-deficient mice are less able to clear experimental *S. typhimurium* challenge at 24 hours is not directly linked to the earlier results mechanistically, although there is abundant evidence of retinoid shaping of immunity that makes this beyond the scope of the paper.

Specific comments:

1) The authors nicely show that intestinal and liver SAA expression is reduced following a vitamin A deficient diet. To support a direct effect, do the authors know if Vitamin A supplementation reverses these changes?

2) The list of other transcripts in the list presented in Figure 1—figure supplement 1 is intriguing and it would be helpful if the authors could comment on this a little more. H-Antigen, dual oxidases, the NHE antiporter and the granzymes all have interesting disease associations.

3) In Figure 1 the open bars for IL-1beta + IL-6 without retinol appear to be identical for both SAA1 and SAA2: this may be coincidence, but I wondered whether an error had crept in?

4) Can the authors add any further information on how retinol/RA induces SAA in HepG2 cells?

5) Further information on the route of infection is required for Figure 7. What is the impact of SAA deficiency on the intestinal response to *S. typhimurium*? It would also be better to show the geometric means in view of the log scale.

6) What role might the constitutive (SAA5) play in the process of retinol transport? Given the role of SAA as an acute phase protein, what are the consequences of induction during non-infectious inflammation (e.g. autoinflammatory syndromes) from retinoid signaling?

---

## [Author Response]

*The starting point of the paper was an Affy transcript abundance analysis that captured the coincidence of ileal gene expression alterations comparing germ-free and colonised mice with alterations depending on whether colonised mice were maintained on a vitamin A-deficient or -sufficient diet. The genes that show consistent expression differences include saa 1 and 2 (please note that I assume that saa 3 should be substituted for one of the saa1 labels, which appear twice in*
Figure 1—figure supplement 1*)*.

The figure is correct as labeled. There are multiple probe sets representing *Saa1* on the Affymetrix arrays and both of these probe sets showed a change in transcript abundance. The change in *Saa3* transcript abundance that was detected by Q-PCR analysis in Figure 1 was not captured by the microarray for reasons that are unclear.

*Specific comments*:

*1) The authors nicely show that intestinal and liver SAA expression is reduced following a vitamin A deficient diet. To support a direct effect, do the authors know if Vitamin A supplementation reverses these changes*?

This is a great question, and we have performed additional experiments to address this point. To test whether retinoids directly impact intestinal *Saa* expression we performed experiments with intestinal explants supplemented with retinol directly delivered onto the mucosal surface. We initially tried this experiment with explants from vitamin A-deficient mice but found that the tissue viability was extremely poor. However, we were able to detect increased *Saa1* and *2* expression in small intestinal explants harvested from vitamin A-replete mice that were supplemented with direct mucosal delivery of retinol. (While we can detect *Saa3* in vivo, we were unable to detect it in our explant system, likely due to its substantially lower levels of expression relative to *Saa1* and *2*.) These results are now shown in the new Figure 1–figure supplement 3.

(Note that oral supplementation of retinoids is problematic as retinoids are notoriously labile and thus readily degraded by the acidic pH of the stomach.)

To assess liver *Saa* expression in retinoid supplemented mice, we performed in vivo reconstitution experiments where retinoic acid was administered intraperitoneally to vitamin A-deficient mice for 3 days. This treatment resulted in elevated expression of both *Saa1* and *Saa2* in the livers of supplemented mice relative to vitamin A-deficient controls. These data are now included in new Figure 1–figure supplement 3.

Together, these data complement our cell culture analysis in HepG2 cells (Figure 1) and support the idea that retinoids directly impact SAA expression

*2) The list of other transcripts in the list presented in*
Figure 1—figure supplement 1
*is intriguing and it would be helpful if the authors could comment on this a little more. H-Antigen, dual oxidases, the NHE antiporter and the granzymes all have interesting disease associations*.

Several of the other transcripts identified in the microarray analysis in Figure 1—figure supplement 1 are indeed interesting and represent targets for future analysis of how vitamin A impacts mucosal function. There is a known role for retinoic acid in CD8^+^ T cell differentiation (Allie et al., J. Immunol*.*, vol. 190, p. 2178-2187), which could explain the lowered abundance of granzyme transcripts (GzmA and GzmB) in the vitamin A-deficient mice. More interestingly, the array unexpectedly revealed that expression of three mucus-modifying glycosyltransferases (Fut2, Gcnt1, and B3galt5) is sensitive to dietary vitamin A, suggesting that vitamin A status could impact mucus glycan structure. The potential significance of the increased abundance of sodium:hydrogen antiporter (Slc9a3) transcripts is less apparent as there are no prior reports of its transcriptional regulation by retinoids. We have now added a comment about these findings to the legend of Figure 1—figure supplement 1.

*3) In*
Figure 1
*the open bars for IL-1beta + IL-6 without retinol appear to be identical for both SAA1 and SAA2: this may be coincidence, but I wondered whether an error had crept in*?

We double-checked our data, and this is a coincidence. Further, although the figure shows that the average values are similar, close inspection reveals that the errors are slightly different.

*4) Can the authors add any further information on how retinol/RA induces SAA in HepG2 cells*?

This is an important question and we have continued to push forward our understanding of the molecular basis for retinoid stimulation of SAA expression. We now have an extensive dataset that includes pharmacologic inhibition studies, siRNA knockdown experiments, and *in vivo* genetic experiments, which point to a specific transcriptional mechanism. Given the amount of data that we have accumulated on this point, we are planning to prepare a separate manuscript that describes this mechanism in detail, and are hoping that the reviewers agree that this lies outside the scope of the current manuscript.

*5) Further information on the route of infection is required for*
Figure 7.

Thank you for bringing this to our attention. We have now added this information to the figure legend.

*What is the impact of SAA deficiency on the intestinal response to* S. typhimurium*?*

An important point is that the *Saa1/2*^*-/-*^ mice retain expression of SAA3, which is produced in the intestine but not in the liver or bloodstream. This was a key reason for our focus on the response of the *Saa1/2*^*-/-*^ mice to systemic rather than oral bacterial challenge. When we challenge the *Saa1/2*^*-/-*^ mice orally with *S. typhimurium*, we detect no statistically significant differences in luminal colonization levels or dissemination to mucosal or systemic sites, as might be predicted by the presence of intestinal SAA3 in these mice. Ongoing work is focused on generating mice that lack all SAA isoforms, as these will likely produce better insight into the role of SAAs in immunity to mucosal infections.

*It would also be better to show the geometric means in view of the log scale*.

We agree, and now show the geometric means in Figure 7.

*6) What role might the constitutive (SAA5) play in the process of retinol transport*?

We thank the reviewers for raising this important point. SAA5 (now designated as mouse SAA4, and equivalent to SAA4 in humans) is expressed in the livers of uninfected mice and humans and secreted into the bloodstream at concentrations of ∼50 ug/ml. This is well below the concentrations of >1 mg/ml that are observed for SAA1 and SAA2 after acute infection, but similar to the serum concentrations of retinol binding protein in healthy individuals. SAA4 shows 54-56% homology to SAA1, 2, and 3, so it is more distantly related. However, it retains the hydrophobic amino acids that are predicted to line the hydrophobic binding pocket in SAA3. Further, we performed homology modeling using the mouse SAA3 structure and the mouse SAA4 sequence that yielded a SAA4 model with a highly similar predicted structure. We therefore predict that SAA4 is also a retinol binding protein. If so, it is possible that SAA4 functions in retinol transport in healthy, non-infected animals. We have added additional text discussing this point to the Discussion section.

*Given the role of SAA as an acute phase protein*, *what are the consequences of induction during non-infectious inflammation (e.g. autoinflammatory syndromes) from retinoid signaling?*

Liver SAA expression is upregulated by cytokines such as TNFα, IL-1β, and IL-6, and thus could also be induced during inflammatory states that are independent of infection. Supporting this idea, elevated serum SAA levels have been observed in patients with rheumatoid arthritis (e.g., Arthritis Res 2, 142-144). We speculate that SAAs mediate retinol transport during such inflammatory states just as they do during infection, and plan to investigate this possibility in our future studies.